# Improving LLM Group Fairness on Tabular Data via In-Context Learning

**Valeriia Cherepanova**
Amazon AWS AI

**Chia-Jung Lee**
Amazon AWS AI

**Nil-Jana Akpinar**
Amazon AWS AI

**Riccardo Fogliato**
Amazon AWS AI

**Martin Andres Bertran**
Amazon AWS AI

**Michael Kearns**
University of Pennsylvania
Amazon AWS AI

**James Zou**
Stanford University
Amazon AWS AI

## Abstract

Large language models (LLMs) have been shown to be effective on tabular prediction tasks in the low-data regime, leveraging their internal knowledge and ability to learn from instructions and examples. However, LLMs can fail to generate predictions that satisfy group fairness, that is, produce equitable outcomes across groups. Critically, conventional debiasing approaches for natural language tasks do not directly translate to mitigating group unfairness in tabular settings. In this work, we systematically investigate four empirical approaches to improve group fairness of LLM predictions on tabular datasets, including fair prompt optimization, soft prompt tuning, strategic selection of few-shot examples, and self-refining predictions via chain-of-thought reasoning. Through experiments on four tabular datasets using both open-source and proprietary LLMs, we show the effectiveness of these methods in enhancing demographic parity while maintaining high overall performance. Our analysis provides actionable insights for practitioners in selecting the most suitable approach based on their specific requirements and constraints.

## 1   Introduction

In recent years, the scope of large language models (LLMs) has broadened significantly beyond traditional natural language processing tasks, with recent research demonstrating their effectiveness in tackling challenges on tabular data, including predictive tasks [1, 2]. Typically, structured data is converted into textual format and provided to the language model along with a concise task description and key features. Notably, it has been shown that language models are particularly beneficial in scenarios with limited training data, as they can utilize internal knowledge about world from pre-training combined with textual instructions and few-shot examples to make predictions [3].

Although considerable research has been devoted to exploring and addressing issues of stereotypical bias and fairness in language models applied to natural language tasks, tabular datasets present distinct challenges, particularly in group fairness. It is important to differentiate group fairness in the context of tabular data from conventional notions of fairness in NLP tasks: group fairness in tabular problems hinges on class labels and the representation of various demographic groups within these labels, while stereotypical fairness in NLP has primarily focused on bias in model representations. Notably, achieving fairness in the typical NLP sense does not automatically ensure group-fair predictions in tabular tasks due to potential disparities in class distributions.

Recent studies have started exploring how language models handle group fairness when applied to tabular data, revealing noticeable fairness discrepancies among different demographic groups. [4] and [5] evaluate a few baseline methods for improving group fairness in tabular tasks, including

SafeGenAI Workshop @ 38th Conference on Neural Information Processing Systems (NeurIPS 2024).

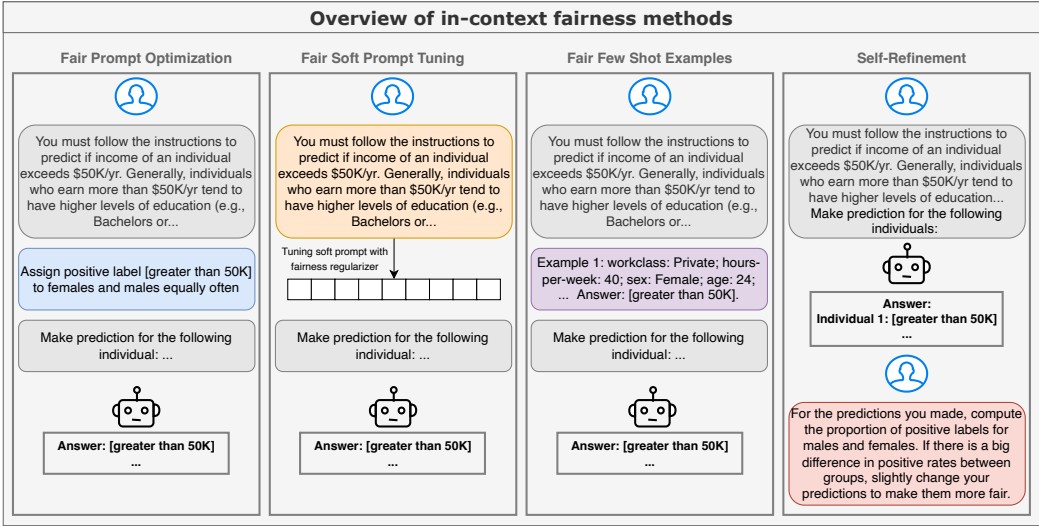

Figure 1: **Overview of fairness methods explored in this work.** We focus on in-context learning approaches, including fair prompt optimization and soft prompt tuning, fair few-shot examples, and self-refinement. For each method, we highlight the specific prompt components optimized in these approaches using different colors, while components of the prompts highlighted in gray do not change across strategies.

resampled fine-tuning, and few-shot learning with label flipping and find these methods to have limited effectiveness. A recent survey paper [6] recognizes the challenge of mitigating inherent biases in large language models through conventional fine-tuning and few-shot learning and highlights the need for more effective strategies to address group unfairness in tabular tasks.

In this work we examine four approaches for empirically improving demographic parity of LLMs when applied to making predictions on tabular datasets. These approaches include in-context methods such as prompt optimization, soft prompt tuning, few-shot in-context learning, and self-refining predictions to promote fairness. We empirically evaluate these methods using both open-source and proprietary models across four tabular datasets, demonstrating their effectiveness. Based on our analysis, we provide actionable recommendations to practitioners on the most suitable method for different scenarios, and discuss how these approaches may be adapted to other notions of fairness.

## 2 Related Work

### 2.1 Large Language Models on Tabular Data

A growing body of work has applied deep learning algorithms to tabular data [2, 7, 8, 9, 10]. Relevant to our setting, some of these studies have employed LLMs to analyze tabular data that is serialized into formatted text. They show that descriptive feature names, well-defined instructions, in-context examples, and chain-of-thought reasoning enhances LLM performance [11, 12, 13]. Some specifically focus on classification tasks [1, 14, 15, 6, 16], which is also the focus of our work. The prior knowledge of LLMs allows them to perform better than traditional algorithms such as XGBoost in low-data regimes [3, 1]. However, LLM predictions can reflect inherent biases, affecting the fairness of their outcomes [17, 18]. [18] is closely related to our work: they analyze the accuracy and fairness of LLM predictions, concluding that traditional ML models exhibit fewer disparities. Although in-context learning and finetuning do not fully close the fairness gap, label-flipping in in-context examples significantly reduces biases, albeit at the cost of prediction performance. Our work contributes to this literature by introducing four in-context learning approaches for mitigating the demographic parity gap in tabular data predictions, demonstrating their effectiveness across widely-used fairness datasets.

## 2.2 Bias and Stereotypes in LLMs

Despite their promising capabilities, language models also exhibit biases and stereotypes [19, 20, 21]. These biases mostly originate from the training data, which often contain historical and societal prejudices embedded within the text. Biases have been reported with respect to several demographic groups, e.g., gender, race, ethnicity, and socioeconomic status [22, 23, 24]. With the use of these models becoming more widespread, these biases have the risk to substantially reinforce harmful stereotypes and perpetuate existing inequalities, especially when deployed in high-stakes settings [25]. Addressing these biases is essential, and several mitigation strategies have been proposed for this purpose, including data augmentation and prompt tuning [26, 27, 28, 29, 30, 31]. However, effectively applying these strategies to the large-scale pretraining corpora remains challenging. Finally, biases can be hard to detect and several datasets and methods have been proposed to help identify them [32, 33, 34, 35, 36, 37].

## 2.3 Fairness on Tabular Data

Much of the work on classification and algorithmic fairness has focused on tabular datasets [38, 39, 40, 41, 42, 43, 44]. Consequently, there is a wide range of research describing the properties and trade-offs of predictive algorithms on this type of data [45, 46, 47]. Multiple works have proposed fairness-enhancing techniques for traditional ML algorithms (e.g., logistic regression), which generally work by debiasing the data, including a fairness constraint in the optimization problem, or post-processing model predictions [48, 49, 50, 51, 52, 53, 54]. Our work employs related techniques, although some of them are not directly applicable. The formalization of fairness definitions has also been extensively discussed [55]. Fairness metrics evaluated on tabular data typically measure the equality of some target measure across demographic groups, such as accuracy or recall [56], which fall under the umbrella of group fairness definitions (as opposed to individual fairness definitions). One such widely-adopted measure, which we also employ in this work, is demographic parity, which ensures that the frequency of positive predictions is approximately equal across different demographic groups.

## 3 Methods

In this work we consider four empirical approaches for improving group fairness of language model predictions on tabular datasets as illustrated in Figure 1. This section provides a brief overview of each method, with subsequent sections delving into detailed descriptions and experimental results for each approach.

**Fair Prompt Optimization.** We demonstrate the effectiveness of prompt engineering in achieving group fairness in LLMs and show how prompt optimization can be automated. In particular, we propose to optimize a fairness-specific prompt (highlighted in blue on the left panel on Figure 1), appended to the task-specific instructions.

**Soft Prompt Tuning.** In addition to hard prompt optimization, we explore soft prompt tuning, which optimizes the prompt directly in the embedding space instead of discrete token space, see the second-from-left panel in Figure 1. This approach enables direct continuous optimization. We demonstrate the effectiveness of soft prompt tuning with an objective incorporating fairness regularization.

**Fair Few-Shot Examples.** Including class-balanced few-shot examples in-context demonstrated limited effectiveness in previous studies [4, 5]. We instead propose an approach for strategically selecting examples by filtering them based on their similarity to test samples and varying the class label ratios within these examples.

**Self-Refinement.** When making predictions in batches, we can utilize the chain-of-thought and self-refinement capabilities of language models to apply post-hoc corrections to predictions, see the right panel in Figure 1 for an illustration.

# 4   Experimental Details

In our experiments, we focus on scenarios with minimal or no training data, where language models excel by leveraging their inherent knowledge for predictions, often outperforming classical tabular models [1, 3]. For each sample, we prompt the model with task-specific instructions and relevant features (see Appendix C for prompting templates). Optionally, we may also include fairness-specific instructions and few-shot examples, depending on the method used to improve fairness. The answer is then extracted either by generating a response or by calculating token likelihoods for labels.

In experiments involving prompt selection, we use a small validation set of 50 labeled examples to assess model accuracy. We then select Pareto-optimal prompts, which represent those where any improvement in either accuracy or fairness would necessitate a compromise in the other metric. Accuracy is assessed on the validation set, while demographic parity is evaluated on the test set to identify these optimal prompts. We additionally compare our methods against Catboost tabular model trained on 50 examples [57] with fairness constraints applied via Fairlearn's GridSearch function following [58].

**Language Models** We conduct experiments using a variety of widely used language models that vary in size. Due to the computational demands of some methods, we conduct computationally intensive experiments with smaller models and reserve methods that require advanced reasoning for larger language models. Our experiments include Llama 3 8B and 70B [59], Mistral 7B [60], Mixtral 8x7B [61] and Claude Sonnet models [62].

**Datasets** We explore group fairness of LLMs on a set of publicly available datasets widely used in the algorithmic fairness literature. For each of the datasets, we focus on *'gender'* as the protected attribute. The **Adult Income** dataset [63], based on the 1994 US Census, predicts whether an individual's yearly income exceeds \$50k (*1 = yes, 0 = no*). The **German Credit** dataset [64] predicts credit default risk (*1 = good, 0 = bad*) using individual attributes. The **ACS Income & Coverage** data [65], drawn from the US Census, is used for income (*1 = yearly income >\$50k, 0 = else*) and public health coverage (*1 = public health coverage, 0 = else*) prediction tasks, focusing on 2018 data from New York. Additional dataset details are provided in Appendix E.

**Serialization and prompts** LLMs require textual input, unlike traditional tabular prediction models. In line with previous work [3, 1], we serialize data points by (1) mapping categorical values to the respective strings (e.g. *gender = 1* is mapped to *gender = male*), and (2) consolidating column names and entries into one string per row. Although we assume little to no training data, it is reasonable to expect that practitioners will provide task-specific instructions to the model to facilitate accurate predictions. For this, we construct instructions using prototype clustering on the training folds of the datasets, as suggested by [3]. To make instructions more readable, we use GPT-4 to revise prototype information into a single summary paragraph. Please, see Appendix C for more details.

**Metrics** In this work we focus on optimizing *demographic parity* (DP) which aims to equalize positive label selection rate across groups, i.e.
$$\mathbb{E}[f(X) \mid G = g] = \mathbb{E}[f(X)]$$
for a binary predictor $f$ and $g \in \{\text{male}, \text{female}\}$. Constraint violation is reported as ratio between the smallest and largest group level selection rates $\mathbb{E}[f(X) \mid G = g]$ with values closer to $1$ indicating better parity. We use DP primarily because it allows to measure fairness on an unlabeled test set directly and does not require labeled training data. Although our primary focus is on demographic parity, the methods we propose can be adapted to other fairness metrics when labeled training data is available as discussed in section 6. Additionally, while our main objective is demographic parity, we also evaluate *equalized odds* which aims to balance false positive and false negative rates across groups, i.e.
$$\mathbb{E}[f(x) \mid G = g, Y = y] = \mathbb{E}[f(x) \mid Y = y]$$
for a binary predictor $f$, $Y \in \{0, 1\}$, and $g \in \{\text{male}, \text{female}\}$, and report equalized odds ratio between groups.

# 5   Experimental Results

## 5.1   Fair Prompt Optimization

Prompt engineering continues to play an important role in tailoring the capabilities of LLMs to various tasks [66, 67, 68]. Recently, [69] demonstrated that integrating fairness-specific manually-curated

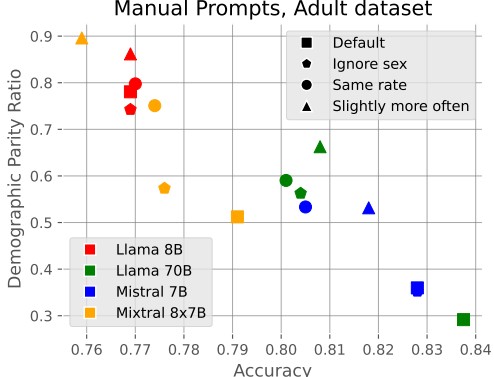
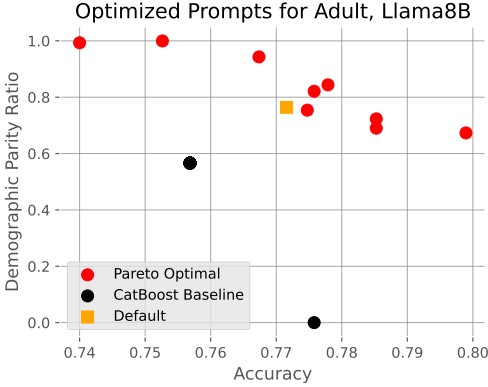

Figure 2: Left: Accuracy and demographic parity for manually constructed fair prompts on Adult dataset, 4 models. Right: Accuracy and demographic parity for fair prompts optimized via a meta-LLM: red points denote Pareto-optimal fair prompts, the orange square shows the default model's performance, and black points depict a CatBoost model optimized on 50 examples with grid search.

instructions in the prompt, such as "it is illegal to discriminate", can attenuate counterfactual biases in model predictions. To evaluate the effectiveness of prompt engineering for fairness, we manually construct a few prompts which are supposed to improve group fairness, such as *(1) "Ignore sex completely when making predictions", (2) "Assign positive label to females and males at the same rate.", (3) "Assign positive label to females slightly more often than to males."*, with the last prompt coming from the fact that female group is the one disadvantaged. We include the accuracy and demographic parity for models queried with these prompts for Adult dataset in Figure 2 left panel.

We observe that, while these prompts can improve demographic parity in some models, finding a universal "fair instruction" that upholds group fairness consistently across multiple models is challenging. Therefore we propose to employ a prompt optimization framework using a meta-LLM, designed to dynamically refine fair instructions.

**LLM as a fair prompt engineer** Several works have shown that LLMs can act as prompt engineers producing performant prompts for downstream models [70, 71]. We adopt this approach following [71] and employ meta-LLM to iteratively refine fairness instructions for the downstream model based on feedback provided from the previous iterations. In particular, we demonstrate the most fair previous instruction and two randomly selected previous instructions, along with their demographic parity scores and selection rates across groups, see example in Figure 3. We refine fairness instructions using the meta-model over 100 iterations. For the meta-LLM we employ the same language model as the one used downstream to make predictions.

In Figure 2 we present the performance of Pareto-optimal fair prompts for the Llama 8B model, along with the performance of a Cat-Boost model baseline. More plots for Mistral and Mixtral models are included in Appendix G. Additionally, Table 1 lists results for the fair prompts which are Pareto-optimal and achieve

---

**Meta-Prompt**

You are a great and creative instruction generator that generates instructions for downstream language models. You must generate a one-sentence "fair instruction" to be appended to an existing general task instruction. The fair instruction should improve demographic parity (equality of selection rates for males and females) in the predictions of the downstream model. The following exemplars show how your fair instruction is used by the downstream model:

You must follow the instructions to predict if income of an individual exceeds $50K/yr.

Generally, individuals who earn more than $50K/yr tend to have higher levels of education (e.g., Bachelors or Prof-school), work in managerial or specialty occupations, and are married with a spouse. They also usually work longer hours per week and have higher capital gains. ... [fair instructions]

Input: workclass: Private; hours per week: 17; sex: Female; age: 42; occupation: Exec-managerial; capital loss: 1485; education: Bachelors; capital gain: 0; marital status: Married-civ-spouse; relationship: Wife;
Output: greater than 50K

The previous instructions, along with their corresponding accuracy, demographic parity, and positive label selection rates across genders, are provided below. You should 1. interpret the performance scores, 2. analyze what should be changed in the fair instruction, and 3. provide new short fair instruction to improve demographic parity, which is very different from the previous ones. Wrap the new fair instruction within < > tags.
The previous instructions are:

Assign positive label [greater than 50K] to females and males with equal probability.
Demographic parity ratio is 0.66
Selection rate for Females is 0.23
Selection rate for Males is 0.35
(...more examples..)

Figure 3: Meta-prompt used to iteratively refine fairness-instructions using a meta-LLM.

Table 1: Performance of optimized fair prompts, tuned soft prompts, and few-shot contexts across 3 models and 4 datasets. We report performance of Pareto-optimal instructions achieving the best validation accuracy and at least 0.9 demographic parity. Bold numbers indicate better accuracy and demographic parity across methods for each model and dataset.

| Model | Adult | | | German Credit | | | ACS Coverage | | | ACS Income | | |
| --- | --- | --- | --- | --- | --- | --- | --- | --- | --- | --- | --- | --- |
| | Acc | DP | EO | Acc | DP | EO | Acc | DP | EO | Acc | DP | EO |
| Catboost + GS | 0.76 | 0.57 | 0.67 | 0.66 | 0.75 | 0.44 | 0.63 | 0.81 | 0.65 | 0.75 | 0.85 | 0.88 |
| Llama8B Default | **0.77** | 0.78 | 0.9 | 0.56 | 0.8 | 0.66 | 0.62 | 0.73 | 0.60 | 0.71 | 0.88 | 0.95 |
| Llama8B+FairPrompt | **0.77** | **0.94** | 0.79 | 0.57 | 0.95 | 0.81 | **0.67** | 0.96 | 0.93 | **0.74** | **0.92** | 0.82 |
| Llama8B+Few-Shot | 0.76 | **0.94** | 0.77 | 0.63 | 0.91 | 0.85 | 0.61 | 0.96 | 0.9 | 0.73 | 0.9 | 0.91 |
| Llama8B+SoftPrompt | 0.73 | **0.94** | 0.84 | **0.66** | **0.97** | 0.90 | 0.59 | **0.97** | 0.88 | 0.69 | 0.89 | 0.92 |
| Mistral7B Default | **0.83** | 0.36 | 0.43 | 0.62 | 0.82 | 0.73 | **0.67** | 0.49 | 0.26 | **0.76** | 0.86 | 0.89 |
| Mistral7B+FairPrompt | 0.81 | 0.55 | 0.77 | **0.7** | 0.9 | 0.68 | 0.66 | **0.94** | 0.88 | 0.71 | 0.92 | 0.91 |
| Mistral7B+Few-Shot | 0.80 | **0.93** | 0.68 | 0.57 | 0.95 | 0.92 | 0.66 | 0.93 | 0.83 | **0.76** | **0.99** | 0.59 |
| Mistral7B+SoftPrompt | 0.75 | 0.90 | 0.62 | 0.65 | **0.97** | 0.90 | 0.56 | 0.92 | 0.88 | 0.75 | 0.85 | 0.91 |
| Mixtral8x7B Default | **0.79** | 0.51 | 0.57 | 0.47 | 0.72 | 0.65 | **0.65** | 0.83 | 0.75 | 0.71 | 0.85 | 0.96 |
| Mixtral8x7B+FairPrompt | 0.78 | **0.95** | 0.61 | **0.58** | 0.94 | 0.83 | 0.64 | **0.99** | 0.9 | 0.72 | 0.92 | 0.89 |
| Mixtral8x7B+FewShot | 0.76 | 0.91 | 0.81 | 0.46 | **0.97** | 0.75 | 0.64 | **0.93** | 0.86 | **0.73** | **0.93** | 0.76 |

at least 0.9 demographic parity ratio. We observe, that these engineered fair prompts significantly improve fairness of the models, often without sacrificing much accuracy. In Appendix G we provide the optimized prompts achieving the best and the worst demographic parity.

## 5.2 Soft Prompt Tuning

In traditional methods, standard in-processing fairness interventions often involve training machine learning models with a fairness penalty. This encourages the model to equalize selection rates or, depending on the penalty, the error rates across demographic groups [48, 52]. Drawing inspiration from these techniques and parameter-efficient fine-tuning methods, we propose a similar approach that can be applied to improving group fairness in language models. In particular, rather than optimizing fair prompts in the discrete space of tokens, as done in the previous section, we suggest optimizing a soft prompt by fine-tuning tokens in the embedding space. Continuous optimization in the embedding space allows us to incorporate the fairness penalty into objective directly. Specifically, we fine-tune 50 tokens initialized with task-specific instructions in the embedding space for 20 epochs. This approach applies a penalty designed to equalize the likelihoods of tokens corresponding to positive labels across groups within a batch:

$$|P(Y = 1|A = 0) - P(Y = 1|A = 1)|.$$

To tune the prompt we use 1000 samples with pseudo-labels obtained by the same language model in the zero-shot setup, simulating a scenario without labeled data. Similarly to our fair prompt engineering experiments, we identify Pareto-optimal points among fine-tuning epochs and include results for Pareto-optimal soft prompts achieving at least 0.9 test demographic parity in Table 1. We observe that while tuning soft prompts improves demographic parity across all datasets, it results in suboptimal trade-off with accuracy compared to hard prompt optimization approach. This could potentially be attributed to the sensitivity of the tuning procedure to hyperparameters or the reliance on pseudo-labels. Additionally, we include plots illustrating fairness-accuracy tradeoff for Pareto-optimal soft prompts in Appendix G.

## 5.3 Fair Few-shot Examples

Prior work [4, 17, 5] has leveraged the in-context learning capabilities of language models for this problem space. They hypothesize that, when selected appropriately, few-shot examples can effectively influence the final predictions to more accurately reflect the desired notion of fairness.

For instance, it has been demonstrated that flipping the labels of few-shot examples can effectively reduce bias, albeit at the expense of significantly lower classification performance [4], while class- and group- balanced selection does not mitigate the bias [5].

Table 2: Results for self-refining approach across three models and four datasets. Bold numbers indicate better demographic parity between the original and refined predictions.

| Model | Adult | | | German Credit | | | ACS Coverage | | | ACS Income | | |
|---|---|---|---|---|---|---|---|---|---|---|---|---|
| | Acc | DP | EO | Acc | DP | EO | Acc | DP | EO | Acc | DP | EO |
| Catboost + GS | 0.76 | 0.57 | 0.67 | 0.66 | 0.75 | 0.44 | 0.63 | 0.81 | 0.65 | 0.75 | 0.85 | 0.88 |
| Llama70B Default | 0.78 | 0.53 | 0.59 | 0.58 | 0.85 | 0.61 | 0.61 | **0.81** | 0.69 | 0.75 | 0.84 | 0.99 |
| Llama70B+Self-Refine | 0.71 | **0.89** | 0.89 | 0.56 | **0.92** | 0.78 | 0.6 | 0.76 | 0.64 | 0.74 | **0.87** | 0.92 |
| Claude Default | 0.79 | 0.50 | 0.57 | 0.66 | 0.93 | 0.89 | 0.63 | **0.77** | 0.64 | 0.76 | 0.82 | 0.94 |
| Claude+Self-Refine | 0.73 | **0.98** | 0.74 | 0.63 | **0.97** | 0.66 | 0.64 | 0.72 | 0.61 | 0.75 | **0.9** | 0.88 |

With a similar goal, we propose a strategy for constructing fair few-shot examples, which differs from the previous methods in three ways. First, instead of randomly sampling examples from the training data, we apply the nearest neighbor search to select examples that are most similar to a current test instance in the feature space[1]. Also, we always select examples that share sensitive attribute with the test instance. Secondly, as we assume no access to training data labels, we use the language models' default zero-shot predictions as pseudo labels to construct demonstrations (similarly to soft prompt tuning experiments). Finally, we extensively manipulate the distributions of positive and negative in-context examples between groups. In particular, we test varying ratios of positive examples for female test samples, $p_f = [0.1, 0.3, 0.5, 0.7, 0.9, 1.0]$, and for male test samples $p_m = [0.1, 0.3, 0.5, 0.7, 0.9, 1.0]$, resulting in 36 ratio pairs. We hypothesise that increasing the number of positive examples for the minority group increases their selection rate, thereby promoting better parity with the majority group.

Figure 4 illustrates the impact of varying the ratio of positive examples in the prompt. The x-axis represents the ratio of positive examples for female test instances, while the color indicates the ratio of positive examples used for predictions on male samples. The results are averaged across 3 random seeds, with the band indicating the standard deviation across seeds. We observe that increasing the positive ratio for females significantly improves demographic parity, to the extent that the selection rate for females surpasses that for males. Additional figures for other models and datasets are displayed in Appendix G. These results confirm that adjusting the ratio of positive examples in-context is an effective method for manipulating the prevalence of positive class predictions, and employing different ratios across protected groups can effectively reduce disparities in selection rates.

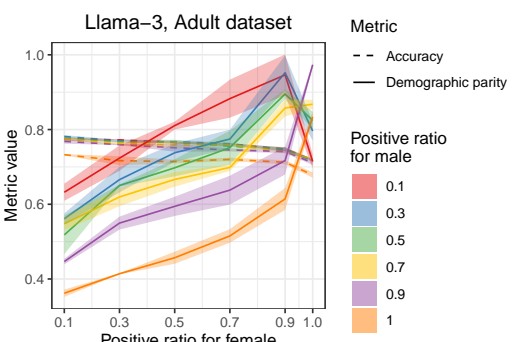

Figure 4: Accuracy and demographic parity ratio metrics for prompts containing few-shot examples with varying number of positive examples across groups, evaluated on Adult dataset using Llama8B.

Additionally, we compare our nearest-neighbor selection strategy with a baseline selecting examples randomly while preserving similar label ratios in-context. Appendix Figure 5 shows that including random in-context examples results in lower demographic parity with larger variance. Also, unlike the nearest-neighbor approach, there is no apparent trend showing that including more positive samples boosts the selection rate for any demographic group, highlighting the importance to including only relevant examples in-context. Finally, in Table 1 we report demographic parity ratio, equalized odds ratio and accuracy metrics for the Pareto-optimal combination of positive label ratios, which achieves the best validation accuracy and at least 0.9 demographic parity.

## 5.4 Self-Refinement

In addition to in-processing methods, fairness literature also includes a wide array of post-processing techniques [52]. These methods work by altering model outputs directly. We propose an LLM-based

---

[1]To compute similarity scores between instances, we use the Jaccard metric as most features are discrete or categorical.

post-processing method that leverages the reasoning capabilities of language models, along with a chain-of-thought process, to refine their own predictions. The self-refinement approach involves using language models to identify individuals from both minority and majority groups who are near the "decision boundary", and then flipping their labels to achieve the desired demographic parity ratio. Therefore, the prediction process includes two stages:

1. The model makes initial predictions on a batch of data samples.
2. The model then assesses demographic parity in a batch and adjusts predictions to attain the desired parity, if necessary.

An example prompt used to refine predictions is illustrated in Figure 1 most right panel. Given that self-refinement approach relies on the advanced reasoning capabilities of language models to analyze predictions, compute metrics of interest, and adjust individual predictions, we conduct these experiments with larger models, specifically Llama3 70B and Claude Sonnet models. We make predictions on a batch of 40 samples, and instruct the model to make adjustments only when the difference in positive rates across groups exceeds 15%. We report the results of the self-refinement approach in Table 2. For all models, refined predictions achieve improved demographic parity across all datasets except for ACS coverage, although this sometimes leads to a notable trade-off in accuracy. In addition, there is no guarantee for similar individuals to receive similar predictions with this method because of the 'correction step' which is at odds with notions of individual fairness [49].

## 6 Discussion

We systematically explore four empirical methods to improve group fairness of language model predictions on tabular datasets. Our experiments across four tabular datasets using multiple language models demonstrate these approaches effectively mitigate demographic disparities. We discuss the key takeaways for each method below.

**Fair Prompt Optimization** can improve not only fairness but also classification performance, contingent upon the model's "creativity." This method involves an optimization process that requires evaluating the prompt on a dataset for a number of iterations. Although the resulting instructions are interpretable, the reasons why specific instructions yield fairer results are not always clear.

**Soft prompt tuning** is computationally expensive and sensitive to the choice of hyperparameters. While this method does not yield interpretable instructions, it enables the integration of common fairness regularizers in a differentiable way and may be particularly effective for smaller models.

**Fair Few Shot Examples** is the most interpretable and predictable method, yielding optimal results across models and datasets when an optimal combination of positive examples ratios is selected. However, it uses a longer context window and may be more computationally expensive for larger datasets because of the number of forward passes needed.

**Self-refinement** requires a model with strong reasoning capabilities and does not guarantee similar predictions for similar individuals. However, this method offers a computational advantage for larger models, as predictions are made and adjusted in batches, reducing overall processing time.

We recommend fair few-shot examples and fair prompt optimization as universal approaches achieving the optimal accuracy tradeoff. Soft prompt tuning can potentially adapt smaller models, while self-refinement is useful for scenarios with limited budgets and larger language models.

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

# A Limitations and Potential Risks

Our work has several limitations. Firstly, it exclusively examines in-context approaches and does not address data pre-processing for bias mitigation or post-hoc methods that modify model outputs directly [52]. Additionally, we do not consider model training and fine-tuning strategies other than soft prompt tuning. Finally, we focus on a single notion of fairness, that is demographic parity, since it can be applied in little to no training data regime, the most practical scenario for language models on tabular datasets. However, most of the discussed methods can be adapted to optimize for other fairness notions, such as equalized odds, when labeled training data is available. For example, the prompt optimization procedure can incorporate alternative fairness metrics in the feedback component of the meta-prompt. Soft prompt tuning can adopt differentiable proxy regularizers to enforce desired fairness criteria, and the few-shot examples approach can demonstrate more examples with ground-truth labels to underrepresented groups.

While the methods explored in this work show promise for improving demographic parity of large language models on tabular prediction tasks, there are also several potential risks to consider. Enhancing group fairness may come at the expense of overall predictive performance, and focusing solely on demographic parity could neglect other important fairness criteria. Moreover, optimizing for group fairness along one dimension could inadvertently amplify or introduce other biases not directly measured. As such, practitioners should carefully evaluate the appropriateness and potential pitfalls of these methods for their specific use case before deploying language models for high-stakes decision making on tabular data.

# B Broader Impact

In-context fairness provides an attractive avenue for steering fair outcomes on large language models. By addressing biases directly within model operations, this research can mitigate the propagation of stereotypes and discriminatory behaviors, particularly in sensitive areas such as healthcare and legal decision-making. Ensuring fairness also builds public trust, encouraging broader adoption and integration of AI into daily life, while promoting equitable access to resources and opportunities for all demographic groups.

This paper democratizes access to these methodologies by discussing several alternatives to accomplish this goal, and by providing guidelines on when each of these methods can be applied most effectively.

# C Prompt Details

In this section we provide the prompt templates used in the default setup to obtain predictions from the models. We highlight task instructions in red, and sample for which the prediction is made in blue. When using an optimized fair prompt, if follows the task instructions. For the fair few-shot examples experiments, we provide selected examples before the test sample in the prompt.

## C.1 Prompt Template for Adult dataset

**User:** You must follow the instructions to predict if income of an individual exceeds $50K/yr. Generally, individuals who earn more than $50K/yr tend to have higher levels of education (e.g., Bachelors or Prof-school), work in managerial or specialty occupations, and are married with a spouse. They also usually work longer hours per week and have higher capital gains. On the other hand, those earning less than or equal to $50K/yr typically have lower levels of education (e.g., HS-grad), work in clerical or craft repair occupations, and may be single or married without a spouse. They also usually work fewer hours per week and have lower capital gains. You must provide an explanation and include answer enclosed in square brackets at the beginning of your response.
workclass: Private; hours-per-week: 40; sex: Male; age: 38; occupation: Craft-repair; capital-loss: 0; education: HS-grad; capital-gain: 0; marital-status: Divorced; relationship: Not-in-family.
You must predict if the individual's income exceeds $50K/yr. Answer with one of the following: [less than or equal to 50K] or [greater than 50K].
**Assistant:** Based on the information provided the answer is

### C.2 Prompt Template for German Credit dataset

**User:** You must follow the instructions to predict the credit risk of a German bank customer based on their profile data. Higher credit amounts and longer durations often indicate a 'bad' classification, particularly for loans intended for new cars or high-value purchases. A negative or low checking status (either '<0' or '0<=X<200') is also associated with higher risk. Conversely, 'good' classifications tend to involve moderate or lower credit amounts, shorter loan durations, and are frequently for consumer goods like radios, TVs, or used cars. Surprisingly, having 'no checking' status tends to lean towards a 'good' classification. More people tend to have good credit risk. You must provide an explanation and include answer enclosed in square brackets at the beginning of your response.
age: 36.0; sex: male; job: skilled; housing: own; savings status: no known savings; checking status: 0<=X<200; credit amount: 2181.0; duration: 30.0; purpose: new car.
You must predict the credit risk of a German bank customer. Answer with one of the following: [good] or [bad].
**Assistant:** Based on the information provided the answer is

### C.3 Prompt Template for ACS Income dataset

**User:** You must follow the instructions to predict if income of an individual exceeds \$50K/yr. Individuals who are most likely have income less than or equal to 50K are typically younger, unmarried, with lower levels of education (high school diploma or some college), and engaged in lower-skilled occupations such as cashiers, janitors, or administrative assistants. Conversely, individuals who are older, married, with higher levels of education (bachelor's or master's degree), and are employed in professions such as teachers, managers, physicians, or registered nurses are more likely to have income greater than 50K. You must provide an explanation and include answer enclosed in square brackets at the beginning of your response.
Sex: Female; Race: White alone; Educational attainment: Bachelor's degree; Relationship: Roomer or boarder; Age: 35.0; Usual hours worked per week past 12 months: 32.0; Marital status: Never married; Place of birth (Recode): Florida/FL; Class of worker: Employee of a private not-for-profit, tax-exempt, or charitable organization; Occupation: EDU-Elementary And Middle School Teachers.
You must predict if the individual's income exceeds \$50K/yr. Answer with one of the following: [less than or equal to 50K] or [greater than 50K].
**Assistant:** Based on the information provided the answer is

### C.4 Prompt Template for ACS Coverage dataset

**User:** You must follow the instructions to predict whether an individual is covered by public health insurance. Individuals covered by public health insurance tend to have a regular high school diploma, have never served in the military, and generally have lower income. In contrast, features such as being employed, having educational attainment, higher income (above \$20,000) and being married correlate with not being covered by public health insurance. In addition, people with disabilities are more likely to be covered by public health insurance. You must provide an explanation and include answer enclosed in square brackets at the beginning of your response.
Sex: Female; Race: White alone; Educational attainment: Associate's degree; Military service: Never served in the military; Disability recode: Without a disability; Total person's income: 0.0; Marital status: Never married; Employment status recode: Not in Labor Force; Employment status of parents: N/A (not own child of householder, and not child in subfamily); Gave birth to child within the past 12 months: No.
You must predict if the individual is covered by public health insurance. Answer with one of the following: [covered] or [not covered].
**Assistant:** Based on the information provided the answer is

## D  Additional Experimental Details and Hyperparameters

**Hyperparameters for Soft Prompt Tuning**  . In the soft prompt tuning experiments, we fine-tune 50 tokens initialized with the task instructions for 20 epochs. We employ a learning rate of $1e-4$ for Llama 8B models and $1e-5$ for Mistral models, allowing the first three epochs for a warm-up with a linear scheduler. During fine-tuning, we use 1000 train samples with pseudo-labels obtained by using the language model in a zero-shot setup, we apply demographic parity regularization with a penalty

| Dataset | Features | Prediction Target |
|---------|----------|-------------------|
| Adult [63] [CC BY 4.0 license] | workclass, hours per week, gender, age, occupation, capital loss, education, capital gain, marital status, and relationship | Yearly income $\geq 50k$ |
| German credit [64] [CC BY 4.0 license] | age, gender, job, housing, savings status, checking status, credit amount, duration, and purpose | Good / bad credit |
| ACS Income [65] [License] | gender, race, educational attainment, relationship, age, usual hours worked per week past 12 months, marital status, place of birth, class of worker, occupation | Yearly income $\geq 50k$ |
| ACS Coverage [65] [License] | sex, race, educational attainment, military service, disability recode, total person's income, marital status, employment status recode, employment status of parents, gave birth to child within the past 12 months | Public health coverage |

Table 3: Summary of datasets and selected features

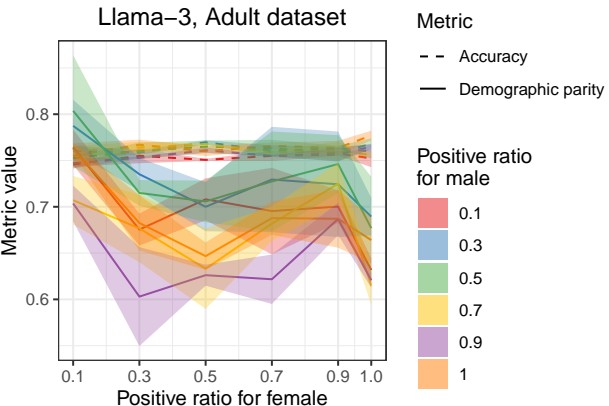

Figure 5: Accuracy and demographic parity ratio metrics for randomly chosen in-context examples with varying number of positive examples across groups, evaluated on Adult dataset using Llama8B.

weight of $0.5$. We employ a class-balanced sampler and set the batch size to 60 samples for Mistral and 50 samples for Llama models, which were the largest sizes we could use given the computational constraints.

# E   Datasets

We include details on the datasets and features used in our experiments in the Table 3.

# F   Hardware

We conducted all experiments using 8 Tesla V100 32GB GPUs through AWS. The soft-prompt tuning experiments required approximately 120 GPU hours per model per dataset, resulting in 950 GPU hours in total. The prompt optimization experiments consumed around 35 GPU hours per model per dataset, resulting in 420 GPU hours for three models and four datasets. Fair few-shot examples experiments took approximately 60 GPU hours per model per dataset for one seed, resulting in 2160 GPU hours of experiments.

# G  Additional Results

## G.1  Additional Results for Fair Prompt Optimization

In Tables 4, 5 we include optimized fair prompts for each dataset each model. In particular, we include Pareto-optimal prompts, which achieve the highest and the lowest demographic parity ratio.

## G.2  Additional Results for Fair Few-Shot Examples

In Figures 6 and 7, we demonstrate accuracy and demographic parity metrics for the prompts containing different proportions of positive and negative few-shot examples across demographic groups. We observe that for all datasets and models, increasing the proportion of positive examples for a demographic group results in a higher selection rate in that group. Additionally, Figure 5 illustrates the trend in demographic parity for prompts including random examples instead of nearest neighbors. In contrast to our strategy, including random examples does not significantly influence the models' selection rates.

## G.3  Comparing Pareto Frontiers

Figure 8 illustrates the Pareto frontiers for prompt optimization, soft prompt tuning, and fair few-shot examples methods. Specifically, it plots the Pareto-optimal prompts for each method, demonstrating the trade-offs between accuracy and fairness metrics for each method.

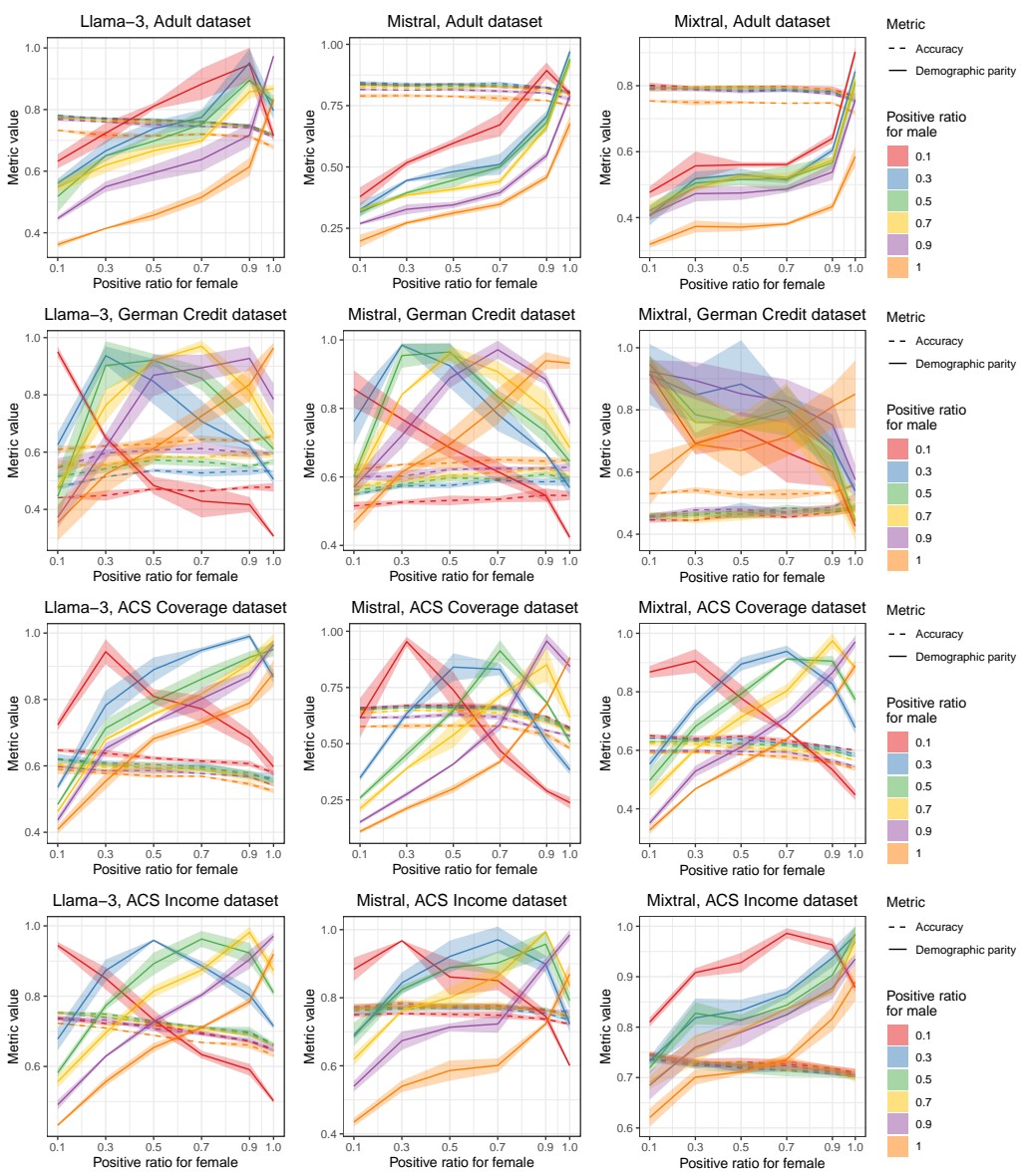

Figure 6: Accuracy and demographic parity metrics for fair in-context fewshots examples.

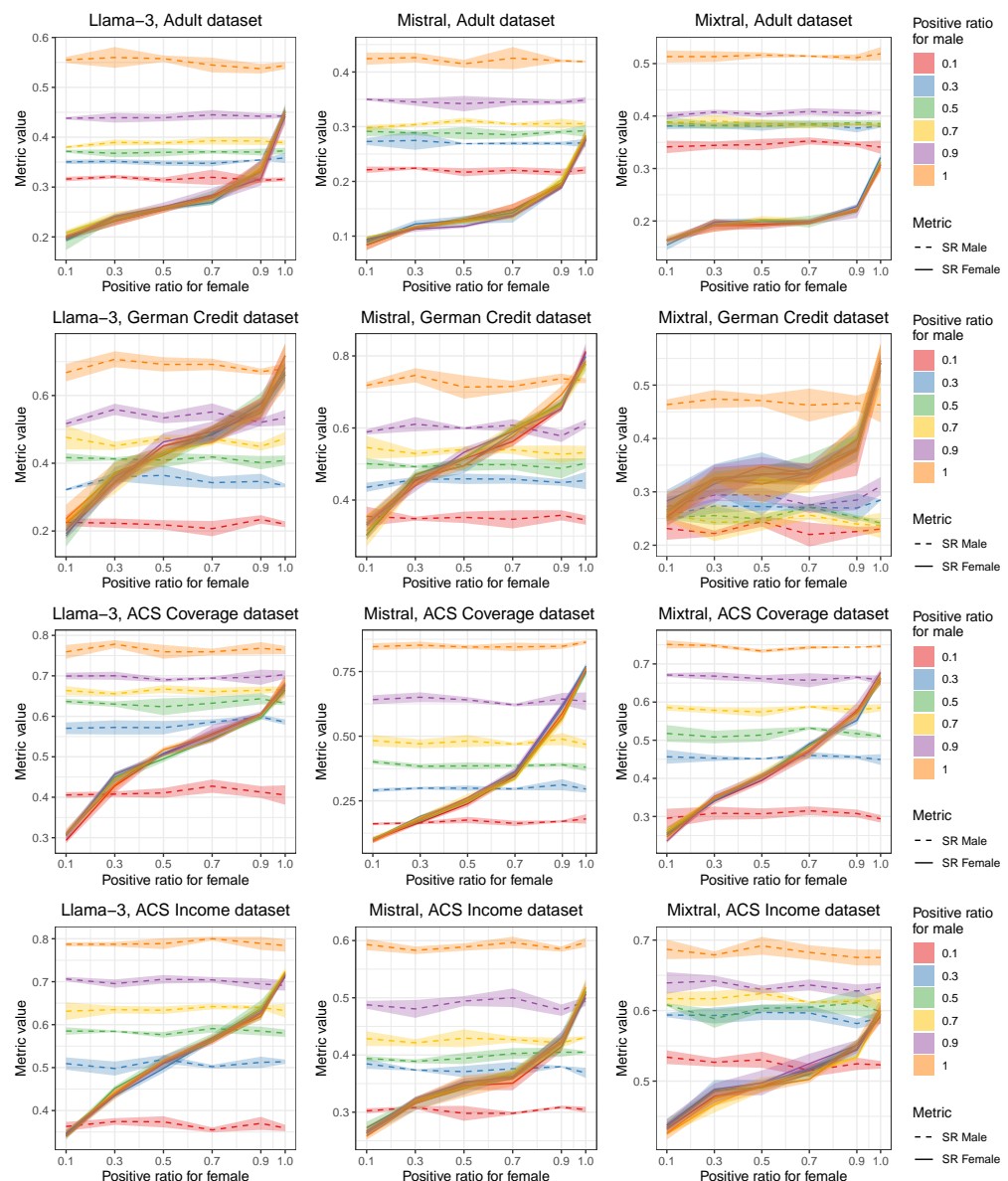

Figure 7: Selection rates for male and female for fair in-context fewshots examples.

| Dataset | Model | Most fair prompt | Least fair prompt |
|---|---|---|---|
| Adult | Llama 8B | Consider the relationship between marital status and income, and ensure that the selection rates for females and males are balanced across different marital statuses, taking into account the potential impact of marriage on income. ($DPR = 0.999$) | Ensure that the selection rates for females and males are balanced across different levels of education (HS-grad, Some-college, Bachelors, etc.) ($DPR = 0.673$) |
| Adult | Mistral 7B | Predict based on features that are statistically independent of both income and gender. ($DPR = 0.553$) | Predict with equal probability for males and females, regardless of other features. ($DPR = 0.308$) |
| Adult | Mixtral 8x7B | Predict the income of an individual, ensuring an equal selection rate for both females and males by accounting for gender parity and other relevant features without bias. ($DPR = 0.994$) | Predict the income of an individual without gender bias, ensuring an equal probability of assigning the positive label (greater than 50K) to both females and males. ($DPR = 0.546$) |
| German Credit | Llama 8B | Consider the impact of job type on credit risk assessment, especially for females with moderate or lower credit amounts and shorter loan durations, and assign positive label [good] to females with unskilled or skilled jobs at a rate that is at least 10% higher than the rate for males with the same job types. ($DPR = 0.950$) | [empty prompt] ($DPR = 0.779$) |
| German Credit | Mistral 7B | Modify the instructions to consider the effect of demographic factors on credit risk predictions, ensuring that the selection rates for males and females are equal for the 'good' label, while maintaining the existing risk factors and their associations with the 'good' and 'bad' labels, but with a focus on mitigating potential bias towards females in the 'good' label selection, and giving slightly more weight to female applicants with similar risk profiles to males. ($DPR = 0.993$) | Modify the instructions to consider the effect of demographic factors on credit risk predictions, ensuring that the selection rates for males and females are equal for the 'good' label, while maintaining the existing risk factors and their associations with the 'good' and 'bad' labels, but with a focus on mitigating potential bias towards females in the 'good' label selection, and giving slightly more weight to male applicants with similar risk factors to achieve parity. ($DPR = 0.817$) |
| German Credit | Mixtral 8x7B | Assign positive label [good] to both males and females while giving equal consideration to all relevant factors, and further emphasize the importance of unbiased selection by including a statement that encourages considering each applicant as an individual, without any preconceived notions or stereotypes based on their demographic attributes. ($DPR = 0.999$) | Assign positive label [good] to both males and females while giving equal consideration to all relevant factors, including credit amount, loan duration, savings status, job type, housing status, specific purchases, and age, to further enhance demographic parity. ($DPR = 0.518$) |

Table 4: Optimized Fair Prompts for Adult and German Credit datasets. We include Pareto-optimal prompts, which achieve the highest and the lowest demographic parity ratio (DPR)

| Dataset | Model | Most fair prompt | Least fair prompt |
|---|---|---|---|
| ACS Cover-age | Llama 8B | Consider the individual's employment status and adjust the prediction by assigning the positive label [covered] to males who are unemployed or have lower income, and to females who are employed or have higher income, in order to achieve a more balanced selection rate between males and females. ($DPR = 0.998$) | Consider the individual's employment status and adjust the prediction to ensure that males and females with similar income levels and employment status have a similar probability of being assigned the positive label [covered]. ($DPR = 0.409$) |
| ACS Cover-age | Mistral 7B | Consider the sex feature as a neutral feature and ensure that it does not provide an unfair advantage or disadvantage in the prediction process ($DPR = 0.995$) | Consider the sex feature as a protected feature and ensure that it does not negatively impact the selection rate for any gender. ($DPR = 0.819$) |
| ACS Cover-age | Mixtral 8x7B | Assign positive label [covered] to individuals with similar income levels and educational attainment, regardless of sex. ($DPR = 0.999$) | Assign positive label [covered] to individuals with similar income levels, while considering sex as a factor to improve demographic parity. ($DPR = 0.989$) |
| ACS In-come | Llama 8B | Consider the individual's occupation and educational attainment, and also take into account the individual's work experience and job seniority, as these factors can significantly impact income, regardless of gender. ($DPR = 0.922$) | Consider the individual's occupation and educational attainment, and also take into account the number of years of work experience, as females tend to have lower median income due to factors such as part-time work, career breaks, and occupational segregation. This may help to increase the representation of females in the [greater than 50K] category. ($DPR = 0.870$) |
| ACS In-come | Mistral 7B | Assign positive label [greater than 50K] to females with a probability that is 3% higher than that of males, but not more than 1% greater than the demographic parity ratio. ($DPR = 0.9812$) | Assign positive label [greater than 50K] to females with a probability that is 1% higher than that of males. ($DPR = 0.861$) |
| ACS In-come | Mixtral 8x7B | Consider factors such as occupation, education, marital status, and weekly hours worked equally for both genders when predicting income levels, and avoid stereotypes related to caregiving roles and occupational expectations. ($DPR = 0.916$) | Treat occupation as a significant factor for income prediction, while considering gender-neutral job titles and avoiding assumptions about income based on stereotypical gender roles. ($DPR = 0.744$) |

Table 5: Optimized Fair Prompts for ACS datasets. We include Pareto-optimal prompts, which achieve the highest and the lowest demographic parity ratio (DPR)

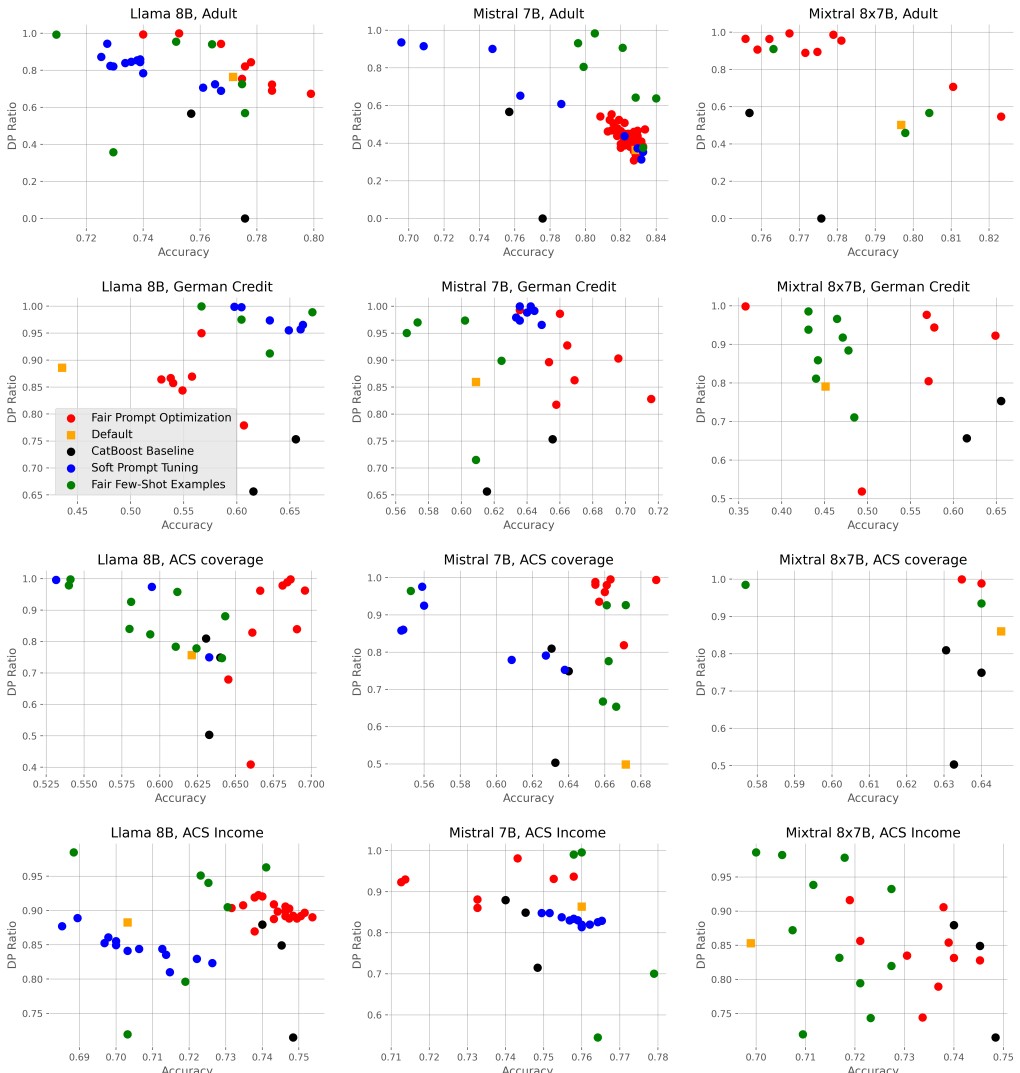

Figure 8: Pareto-optimal points for optimized fair prompts (red), soft prompts (blue) and optimized class ratios for few-shot examples (green) across all datasets and models. Orange square denotes zero-shot performance of the models, while black points correspond to Catboost trained with GridSearch baseline.

