# OpenReview forum: "Improving LLM Group Fairness on Tabular Data via In-Context Learning"
_NeurIPS.cc/2024/Workshop/SafeGenAi — SafeGenAi Poster_

### Official Review · Reviewer_Aqtj · 2024-10-09
**Adequate Experiments and Thorough Analysis.**

**Rating:** 7
**Confidence:** 3

**Review:**

This study explores improving group fairness in large language models (LLMs) for tabular prediction tasks. In this study, the authors propose four methods to enhance fairness: fair prompt optimization, soft prompt tuning, strategic few-shot example selection, and self-refining predictions through chain-of-thought reasoning. Tested on different models, these methods demonstrated improvements in demographic parity while maintaining overall performance.And the authors systematically analyze the potential advantages and disadvantages of these methods.

---

### Official Review · Reviewer_Sjsb · 2024-10-09
**This is a good attempt to improve the fairness of large language models on tabular data, but further analysis is needed.**

**Rating:** 6
**Confidence:** 4

**Review:**

This paper investigates four approaches to improve group fairness, specifically demographic parity, of large language models (LLMs) when applied to tabular data prediction tasks. The methods explored are: (1) fair prompt optimization, (2) soft prompt tuning, (3) strategic selection of few-shot examples, and (4) self-refining predictions via chain-of-thought reasoning. The authors evaluate these methods on four tabular datasets using both open-source and proprietary LLMs. Their results demonstrate the effectiveness of these approaches in enhancing demographic parity while maintaining high overall performance.

Strengths:

- Comprehensive Methodology: The authors systematically investigate four different approaches, providing a thorough exploration of in-context learning methods for improving fairness.
- Empirical Evaluation: The study includes extensive experiments across multiple datasets and LLM architectures, providing robust evidence for the effectiveness of the proposed methods.

Weaknesses:

- Limited Performance and Fairness Metrics: The primary focus on demographic parity, while justified, may not capture the full spectrum of fairness concerns in real-world applications. For performance, other metrics (e.g., precision, recall) and the measurement of each demographic group are also important for analysis.
- Limited Discussion on Dataset: The paper does not thoroughly address how these methods would scale to larger datasets or more complex tabular prediction tasks. At the same time, the authors should explore more real-world datasets with more balanced distributions, rather than datasets where only these sensitive attributes can completely influence the prediction results.
- Potential for Overfitting: The use of a small validation set (50 samples) for prompt selection raises concerns about the generalizability of the optimized prompts.

---

### Official Review · Reviewer_2iXA · 2024-10-11
**Review Comment**

**Rating:** 7
**Confidence:** 3

**Review:**

This paper addresses a critical challenge in the application of Large Language Models (LLMs) to tabular prediction tasks—ensuring group fairness. The authors propose and systematically investigate four empirical approaches to mitigate group unfairness while maintaining high performance. The paper makes several valuable contributions to both fairness research and the practical deployment of LLMs in tabular settings:

Novelty and Relevance: The focus on enhancing group fairness in LLMs for tabular datasets is both timely and highly relevant. While LLMs have demonstrated success in a range of natural language tasks, adapting fairness techniques to tabular data represents a crucial advancement. The paper effectively highlights the limitations of conventional debiasing techniques and proposes a clear, structured approach to overcoming these challenges.

Diverse Fairness Techniques: The investigation of four distinct methods—fair prompt optimization, soft prompt tuning, strategic example selection, and chain-of-thought reasoning—demonstrates the authors' comprehensive approach. The methods are well-suited to the problem at hand, offering versatility for practitioners looking to enhance fairness across various LLMs and datasets.

Effective Experimental Validation: The experiments conducted on four tabular datasets, using both open-source and proprietary LLMs, provide robust validation of the proposed methods. By showcasing improvements in demographic parity without compromising performance, the authors convincingly demonstrate the feasibility of integrating fairness into LLM predictions for tabular tasks.

Practical Guidance: The paper provides actionable insights for practitioners, which is a major strength. By outlining how each approach can be selected based on specific requirements and constraints, the authors make the work directly applicable to real-world use cases. This aspect greatly enhances the paper's practical value for both researchers and industry professionals.

---

### Official Review · Reviewer_VZW6 · 2024-10-11
**Good paper with multiple evaluations of fairness methods for LLM on tabular data**

**Rating:** 7
**Confidence:** 4

**Review:**

### **Summary**:
The paper presents multiple approches to de-bias the LLM in tabular classification task, focusing on the demographic-group fairness. The proposed methods lead to improvement of fairness metrics while maintaining the accuracy in classification.

### **Quality & Clarity**:
Overall the paper is well written and easy to follow. The included examples on prompting and tunning process helps to surface the methodology, and provides sufficient details for reproduction.

### **Significance**:
The paper deals with the fairness of LLM in handling tabular data. Given the increasing popularity and adoption of LLM, the fairness related issue clearly needs more attention. This work provides some insights and foundation for further developments. The experimental results (e.g. Table 1 and 2) clearly showcase the advantage of proposed methods over previous baselines, signifying the effectiveness.

### **Originality**:
The idea of in-context learning for fairness adjustment is somewhat novel for tabular classification task, although the methods (e.g. instruction prompting selection, soft-prompt tuning, reflection) are quite similar to known techniques for LLM, as summarized in [1] and [2].

### **Limitations**:
The variety and scope of testing datasets is too limited and narrow, as it focuses only on income, credict, and insurance coverage. While these datasets are widely used for benchmarking and ease the definition of bias against certain group, simply relying on them alone largely limits the sightfulness and weakens the validality of proposed approaches.

[1] Sahoo, Pranab, et al. "A systematic survey of prompt engineering in large language models: Techniques and applications." arXiv preprint arXiv:2402.07927 (2024).

[2] Schulhoff, Sander, et al. "The Prompt Report: A Systematic Survey of Prompting Techniques." arXiv preprint arXiv:2406.06608 (2024).